# PSO-Merging: Merging Models Based on Particle Swarm Optimization

## Abstract

Model merging has emerged as an efficient strategy for constructing multitask models by integrating the strengths of multiple available expert models, thereby reducing the need to fine-tune a pre-trained model for all the tasks from scratch. Existing data-independent methods struggle with performance limitations due to the lack of data-driven guidance. Data-driven approaches also face key challenges: gradient-based methods are computationally expensive, limiting their practicality for merging large expert models, whereas existing gradient-free methods often fail to achieve satisfactory results within a limited number of optimization steps. To address these limitations, this paper introduces PSO-Merging, a novel data-driven merging method based on the Particle Swarm Optimization (PSO). In this approach, we initialize the particle swarm with a pre-trained model, expert models, and sparsified expert models. We then perform multiple iterations, with the final global best particle serving as the merged model. Experimental results on different language models show that PSO-Merging generally outperforms baseline merging methods, offering a more efficient and scalable solution for model merging.

## 1 Introduction

In recent years, numerous powerful pre-trained Large Language Models (LLMs) have emerged, serving as the foundation for solving various language-related tasks (Brown et al., 2020; Touvron et al., 2023; Jiang et al., 2023; Grattafiori et al., 2024; Chung et al., 2024; DeepSeek-AI et al., 2024). To unlock the abilities of these LLMs on downstream tasks, post-training techniques such as fine-tuning, reinforcement learning with human feedback (RLHF), direct preference optimization (DPO), and Group Relative Policy Optimization (GRPO) are commonly employed (Rafailov et al., 2024; Ouyang et al., 2022; DeepSeek-AI et al., 2025). However, post-training is time-consuming and often requires substantial GPU and data resources.

Constructing a multitask model by performing post-training on the base model is highly resource-intensive. Fortunately, there are numerous open-source expert models available in the community that have undergone post-training on various downstream tasks. Thus, an alternative approach for building a multitask model is to merge post-trained expert models directly (Li et al., 2023a). By merging expert models in the parameter space, the capabilities of multiple downstream tasks can be consolidated into a single model. Model merging offers the benefit of simplicity and efficiency, demanding less data and fewer GPU resources compared to training.

Existing model merging methods can be broadly categorized based on whether they rely on data guidance. Data-independent approaches, which are often simpler, typically involve operations such as scaling, rescaling, pruning, or weighted merging of task vectors while addressing potential conflicts during the integration process (Yadav et al., 2023; Yu et al., 2024a; Ilharco et al., 2023). However, in the absence of data-specific guidance, these methods often struggle to achieve optimal performance due to their limited ability to adapt to the nuances of specific tasks. On the other hand, data-guided methods typically rely on gradient-based calculations to guide the merging process (Matena & Raffel, 2022; Yang et al., 2023). These approaches face significant challenges when applied to scenarios involving a large number of expert models with substantial parameter sizes, as the computational overhead and complexity become prohibitive. To avoid the need for gradient computation, Akiba et al. (2024) propose leveraging the Covariance Matrix Adaptation Evolution

Strategy (CMA-ES) to search for optimal merging weights based on existing methods. While this approach offers a gradient-free, data-driven alternative, it involves sampling in the solution space, where many samples are discarded in each iteration due to low evaluation scores. As a result, it requires a substantial number of iterative steps to achieve satisfactory results, making it inefficient.

In real-world scenarios, data for the target domain is at least available in limited quantities. The ideal model merging method should efficiently and fully utilize this data as guidance, while avoiding extensive computations. The Particle Swarm Optimization (PSO) (Kennedy & Eberhart, 1995) was originally proposed to search for the optimal solution to a target problem. It does not require gradient computation, instead relying on an evaluation function to assess the objective score, which can optionally incorporate data as guidance. This approach minimizes the need for extensive calculations. In each iteration, PSO guides each solution by leveraging information from other solutions, allowing it to more accurately identify the direction toward optimal solutions, which enhances its efficiency. As a result, PSO is particularly well-suited for an ideal model merging method.

In this work, we propose PSO-Merging, a novel model merging method inspired by the traditional PSO. Unlike the traditional approach of randomly initializing the particle swarm in PSO, our method initializes the particle swarm by using each expert model as the starting point. Moreover, we adopt the widely utilized sparsification technique, initially introduced to address parameter conflicts during the merging process. In our method, this technique also enables the generation of a larger number of particles, thereby facilitating a more favorable convergence toward high-quality solutions. After several rounds of the PSO optimization process, we use the final global best particle as the resulting merged model.

We evaluate our method on multiple model architectures, including Flan-T5, LLaMA, and Mistral. Experimental results illustrate that our approach outperforms baseline methods in terms of average scores and achieves significant improvements on certain tasks. Moreover, our experimental analysis demonstrates that PSO-Merging exhibits rapid convergence, and significantly outperforms the baseline methods in merging scenarios involving up to four large expert models.

## 2 METHODOLOGY

In this section, we first give a problem formulation of merging to enhance multitask capability (M-MTC) (Lu et al., 2024), then we introduce PSO-Merging, our novel model merging method based on the PSO. Finally, we provide a brief intuitive explanation of why PSO-Merging works.

### 2.1 PROBLEM FORMULATION

Assume we have a task set $T = \{\tau_1, \tau_2, \cdots, \tau_n\}$ of size $n$. We begin with a pre-trained model parameterized by $\boldsymbol{\theta}_0 \in \mathbb{R}^d$, where $d$ denotes the number of parameters. This model then undergoes post-training on each task in $T$ to adapt and specialize for the specific task. Specifically, for a task $\tau_t$, the pre-trained model is fine-tuned on its corresponding dataset to become an expert in $\tau_t$, parameterized by $\boldsymbol{\theta}_t$. The goal of M-MTC is to merge the set of experts $\boldsymbol{\Theta} = \{\boldsymbol{\theta}_1, \boldsymbol{\theta}_2, \cdots, \boldsymbol{\theta}_n\}$ into a unified model $\boldsymbol{\theta}_{\text{merged}}$ with multitask capability that performs well on $T$.

### 2.2 PSO-MERGING

An overview of PSO-Merging is demonstrated as Figure 1. Our method can be roughly divided into two stages: initialization and iterative updates.

**Initialization** The traditional PSO begins with a randomly initialized solution set $\boldsymbol{\Theta}_{\text{initial}} = \{\boldsymbol{\theta}_1^{(0)}, \boldsymbol{\theta}_2^{(0)}, \cdots, \boldsymbol{\theta}_m^{(0)}\}$ of size $m$. However, at this stage, we initialize the solution set $\Theta_{initial}$ with the original experts along with the sparsified experts, which are acquired by the sparsification mechanism. The sparsification mechanism is widely employed in model merging to mitigate parameter conflicts (Yadav et al., 2023; Yu et al., 2024a; Deep et al., 2024). In our approach, we utilize sparsification not only to address parameter conflicts but also to increase the number of particles (initial solutions). A larger particle pool enables PSO to converge more effectively toward an optimal solution. For simplicity, we adopt the sparsification strategy from DARE. Specifically, for parameters $\boldsymbol{\theta}_t$ and a drop rate $p$, the sparsified parameters $\widetilde{\boldsymbol{\theta}_t}$ are obtained as follows:

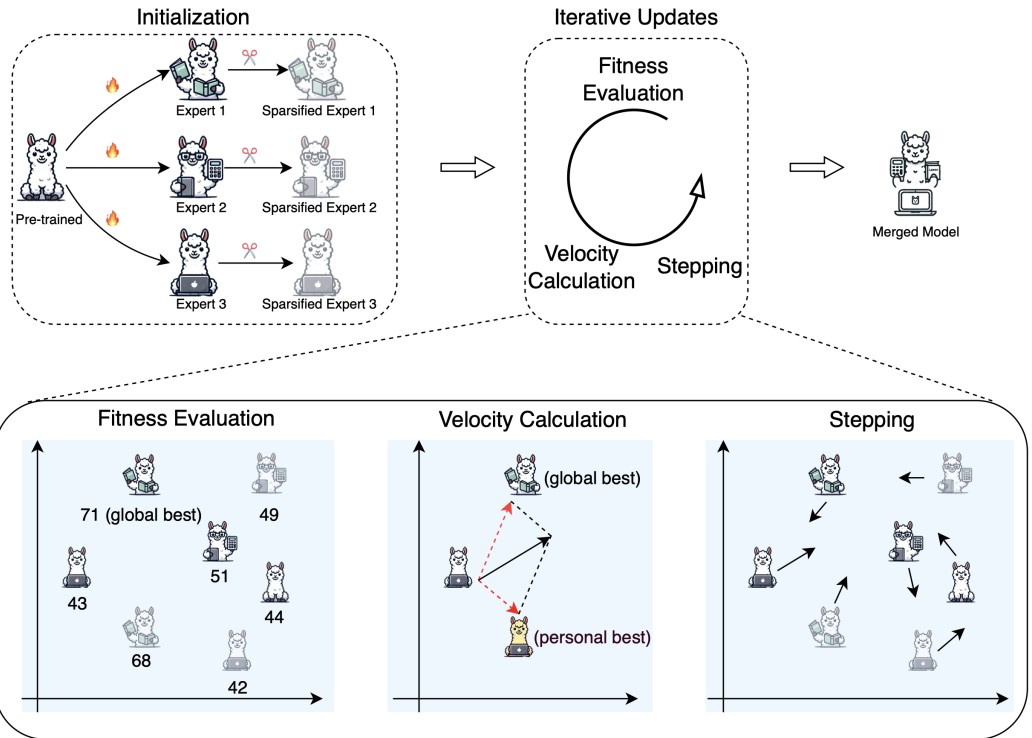

Figure 1: An overview of PSO-Merging. We begin by sparsifying all fine-tuned LLM experts. The swarm consists of the pre-trained model, the fine-tuned LLM experts, and the sparsified fine-tuned LLM experts. The update cycle consists of three steps: fitness evaluation, velocity calculation, and stepping. The axes in the figure represent the parameter space. The numbers in the fitness evaluation present each particle's fitness score, corresponding to the average multitask score in our work. The black arrows indicate the stepping direction of each particle. For simplicity, we omit the momentum term in the velocity calculation.

$$m_i^t \sim \text{Bernoulli}(p), \quad i = 1, 2, \ldots, d, \tag{1}$$

$$\mathbf{m}^t = [m_1^t, m_2^t, \cdots, m_d^t] \in \mathbb{R}^d, \tag{2}$$

$$\widetilde{\boldsymbol{\theta}}_t = (\mathbf{1} - \mathbf{m}^t) \odot (\boldsymbol{\theta}_t - \boldsymbol{\theta}_0)/(1-p) + \boldsymbol{\theta}_0, \tag{3}$$

where $\odot$ represents element-wise multiplication, and $\boldsymbol{\theta}_0$ denotes the pre-trained parameters.

For the expert set $\boldsymbol{\Theta} = \{\boldsymbol{\theta}_1, \boldsymbol{\theta}_2, \cdots, \boldsymbol{\theta}_n\}$, by using the sparsification technique, we acquire the sparsified expert set $\widetilde{\boldsymbol{\Theta}} = \{\widetilde{\boldsymbol{\theta}_1}, \widetilde{\boldsymbol{\theta}_2}, \cdots, \widetilde{\boldsymbol{\theta}_n}\}$. To maximize the use of existing resources to increase the number of particles, we also include the pre-trained model in the initial solutions. So our initial solution set can be represented as

$$\boldsymbol{\Theta}_{\text{initial}} = \boldsymbol{\Theta} \cup \widetilde{\boldsymbol{\Theta}} \cup \{\boldsymbol{\theta}_0\}. \tag{4}$$

**Iterative Updates** In this stage, we perform iterative updates for several steps to approach a good solution. Each update cycle consists of three main steps: fitness evaluation, velocity calculation, and position updating. Traditional Particle Swarm Optimization (PSO) defines the fitness function of a solution as the score of a specific task being solved. However, in our M-MTC scenario, we redefine the fitness function to represent the average score across all tasks, expressed as $f(\boldsymbol{\theta}) = \frac{1}{n} \sum_{i=1}^{n} \text{score}_i(\boldsymbol{\theta})$. The velocity of each solution (similar to a gradient) is determined by its own personal best position and the global best position. The velocity formula of solution $t$ on step $i$ can be represented as:

$$\boldsymbol{v}_t^{(i)} = w \cdot \boldsymbol{v}_t^{(i-1)} + c_1 \cdot r_1 \cdot (\boldsymbol{\theta}_{\text{gbest}}^{(i-1)} - \boldsymbol{\theta}_t^{(i-1)}) + c_2 \cdot r_2 \cdot (\boldsymbol{\theta}_{t,\text{pbest}}^{(i-1)} - \boldsymbol{\theta}_t^{(i-1)}), \tag{5}$$

where $c_1$ and $c_2$ are parameters used to adjust how much PSO concerns the global and personal information. $r_1$ and $r_2$ are random variables that follow the uniform distribution, specifically, $r_1, r_2 \sim U(0,1)$. $w$ is a parameter that controls the momentum of the movement. The personal best position $\boldsymbol{\theta}_{t,\text{pbest}}^{(i)}$ and the global best position $\boldsymbol{\theta}_{\text{gbest}}^{(i)}$ until step $i$ are defined as $\boldsymbol{\theta}_{t,\text{pbest}}^{(i)} = \boldsymbol{\theta}_t^{(\arg\max_{j \leq i} f(\boldsymbol{\theta}_t^{(j)}))}$, $\boldsymbol{\theta}_{\text{gbest}}^{(i)} = \boldsymbol{\theta}_{\arg\max_t f(\boldsymbol{\theta}_{t,\text{pbest}}^{(i)}),\text{pbest}}^{(i)}$.

Then we update each solution according to its own velocity:

$$\boldsymbol{\theta}_t^{(i)} = \boldsymbol{\theta}_t^{(i-1)} + \boldsymbol{v}_t^{(i)}. \tag{6}$$

After iterating for $S$ steps starting with $\boldsymbol{\Theta}_{\text{initial}}$, we choose the final global best particle as our merged LLM, denoted as $\boldsymbol{\theta}_{\text{merged}} = \boldsymbol{\theta}_{\text{gbest}}^{(S)}$.

### 2.3 An Intuitive Explanation for Why PSO-Merging Works

Expanding Equation 6, we obtain the following expression:

$$\boldsymbol{\theta}_t^{(i)} = c_1 \cdot r_1 \cdot \boldsymbol{\theta}_{\text{gbest}}^{(i-1)} + c_2 \cdot r_2 \cdot \boldsymbol{\theta}_{t,\text{pbest}}^{(i-1)} + (1 - c_1 \cdot r_1 - c_2 \cdot r_2) \cdot \boldsymbol{\theta}_t^{(i-1)} + w \cdot \boldsymbol{v}_t^{(i-1)}, \tag{7}$$

where, when ignoring the momentum term $w \cdot \boldsymbol{v}_t^{(i-1)}$, the equation represents a linear combination of $\boldsymbol{\theta}_t^{(i-1)}$, $\boldsymbol{\theta}_{\text{gbest}}^{(i-1)}$ and $\boldsymbol{\theta}_{t,\text{pbest}}^{(i-1)}$. Previous studies have demonstrated the effectiveness of this linear combination (Ilharco et al., 2023). Intuitively, iterating multiple times allows us to find a more optimal linear combination, guided by the data, thereby improving the merged model. The momentum term further helps balance exploration and exploitation by maintaining a particle's velocity, enabling it to escape local optima and enhancing the overall global search capability (Kennedy & Eberhart, 1995).

Earlier studies have shown that the sparsification mechanism can help mitigate parameter conflicts (Yu et al., 2024a). Since our initial particle swarm includes sparsified models (which contribute to the initial states $\theta_t^{(0)}$ and subsequently influence $\theta_{\text{gbest}}^{(i)}$ and $\theta_{t,\text{pbest}}^{(i)}$ throughout iterations), the search process benefits from exploring regions influenced by this initial sparsification. This helps mitigate parameter conflicts when forming the combined parameters $\theta_t^{(i)}$.

## 3 Experiments

In this section, we present the experimental results obtained using four different base language models: Flan-T5-Base (Chung et al., 2024), Llama-3-8B (Grattafiori et al., 2024), Llama-2-13B (Touvron et al., 2023), and Mistral-7B-v0.3 (Jiang et al., 2023). The results demonstrate that our method achieves performance superior to the baseline methods, thereby proving its effectiveness and highlighting its advantages.

### 3.1 Baselines

We choose the following model merging methods as our baseline methods.

**Task Arithmetic** This method involves scaling each task vector by a factor and then combining them with the pre-trained model (Ilharco et al., 2023).

**DARE-Linear** Each task vector is first sparsified randomly and rescaled before being merged into the pre-trained model (Yu et al., 2024a).

**TIES-Merging** This method retains the top-k parameters based on absolute values, resolves sign conflicts among different task vectors, and then integrates them into the pre-trained model (Yadav et al., 2023).

**DARE-TIES**   Task vectors are initially sparsified randomly and rescaled, followed by resolving sign conflicts across task vectors before merging them with the pre-trained model (Yu et al., 2024a; Yadav et al., 2023).

**DELLA-Merging**   Parameters are pruned based on their magnitudes, with different pruning probabilities assigned accordingly. Post-pruning, the process follows the same steps as TIES-Merging (Deep et al., 2024).

**RankMean**   This approach determines merging weights for parameters across expert models based on their relative rank in terms of weight change magnitude. The parameters in each module are then aggregated through a weighted average using these coefficients (Perin et al., 2024).

**Evo**   We adopt the parameter-space merging component of Akiba et al. (2024)'s method. This approach uses CMA-ES to search for optimal merging parameters based on existing methods. We use Task Arithmetic as the base method and employ CMA-ES to explore the merging weights.

**Adamerging**   This method directly treats the merging weights as trainable parameters, optimizing them by minimizing entropy on unlabeled test samples as a surrogate objective function.

**Fisher-Merging**   This approach leverages labeled data from each task to estimate a diagonal approximation of the Fisher matrix, which is then interpreted as the importance of the corresponding task-specific expert.

**RegMean**   This method combines multiple models by minimizing the difference in their predictions on training data, often using inner product matrices.

## 3.2 IMPLEMENTATION DETAILS

We conducted experiments using four distinct base language models: Flan-T5-Base, Llama-2-13B, Llama-3-8B, and Mistral-7B-v0.3. In our approach, we set the parameters $c_1 = 2$, $c_2 = 2$, and $w = 0.2$. We set the total optimization steps $S = 50$ for the Flan-T5-Base experiments and $S = 5$ for other experiments. For the Evo baseline, to ensure a fair comparison, we set the number of evaluation iterations to $n * S$ in all experiments, where $n$ denotes the number of experts and $S$ corresponds to the number of iterations in PSO-Merging. This aligns the evaluation count with that of PSO-Merging. However, in the actual implementation, the number of evaluation iterations in Evo slightly exceeds the set value. For the sparsification component, we applied a drop rate of $p = 0.8$ in all methods that incorporate sparsification including ours. For all baseline methods that include a fixed scaling term, we choose the scaling term to be either $\frac{1}{n}$ or 1.0, where $n$ is the number of expert models. We report the result with the higher average score.

Table 1: Multitask performance when merging experts based on Flan-T5-Base.

| Method | COLA | MNLI | MRPC | QNLI | QQP | RTE | SST2 | STSB | AVG |
|---|---|---|---|---|---|---|---|---|---|
| Task Arithmetic | 69.13 | 62.65 | 79.41 | 89.80 | 83.86 | 81.23 | 91.74 | 73.22 | 78.88 |
| DARE-Linear | 69.51 | 63.79 | 79.66 | 89.88 | **83.89** | 81.23 | 91.74 | 69.83 | 78.69 |
| TIES-Merging | 69.22 | 59.39 | 77.70 | 89.33 | 83.36 | 80.51 | 91.28 | 68.38 | 77.40 |
| DARE-TIES | 69.32 | 62.50 | 79.66 | 89.77 | 83.83 | 81.59 | 91.28 | 71.10 | 78.63 |
| DELLA-Merging | 69.32 | 64.40 | 79.90 | **89.90** | 83.82 | **81.95** | 91.06 | **75.96** | 79.54 |
| Rankmean | 69.13 | 56.45 | 76.23 | 88.45 | 82.12 | 80.14 | 91.17 | 62.21 | 75.74 |
| Evo | **70.85** | 82.91 | 75.74 | 89.35 | 73.91 | 80.87 | **92.20** | 69.70 | 79.44 |
| Adamerging | 69.89 | 77.17 | 79.90 | 89.80 | 81.73 | 79.06 | 91.40 | 66.05 | 79.38 |
| Fisher-Merging | 69.32 | 54.03 | 76.72 | 84.64 | 83.57 | 77.62 | 88.07 | 74.35 | 76.04 |
| RegMean | 69.13 | 26.64 | 75.25 | 79.33 | 77.17 | 61.73 | 86.01 | 48.14 | 65.43 |
| PSO-Merging | 68.17 | **83.80** | **80.64** | 89.53 | 83.56 | 81.23 | 91.06 | 71.94 | **81.24** |

**Flan-T5-Base Experiments**   Following the experimental settings of previous work (Tang et al., 2024), we selected eight text-to-text generation tasks from the General Language Understanding

Table 2: Multitask performance when merging experts based on Llama-2-13B, Llama-3-8B, and Mistral-7B-v0.3.

| | Method | AlpacaEval | MBPP | GSM8K | AVG |
|---|---|---|---|---|---|
| Llama-2-13B | Task Arithmetic | **82.48** | 16.80 | 54.13 | 51.14 |
| | DARE-Linear | 69.36 | 4.00 | 29.80 | 34.38 |
| | TIES-Merging | 64.82 | **32.80** | 59.06 | 52.23 |
| | DARE-TIES | 73.40 | 8.20 | 32.52 | 38.04 |
| | DELLA-Merging | 74.38 | 9.20 | 36.24 | 39.94 |
| | Rankmean | 55.13 | 30.80 | 57.54 | 47.82 |
| | Evo | 61.08 | 32.60 | 56.86 | 50.18 |
| | PSO-Merging | 80.11 | 26.00 | **64.37** | **56.82** |
| Llama-3-8B | Task Arithmetic | 63.79 | 31.00 | 56.56 | 50.45 |
| | DARE-Linear | 60.98 | 8.00 | 53.75 | 40.91 |
| | TIES-Merging | 73.72 | 45.40 | 57.54 | 58.89 |
| | DARE-TIES | 71.69 | 49.20 | **59.97** | 60.29 |
| | DELLA-Merging | 61.22 | 4.60 | 56.18 | 40.67 |
| | Rankmean | 40.96 | 49.40 | 50.72 | 47.03 |
| | Evo | 55.74 | 49.00 | 56.10 | 51.95 |
| | PSO-Merging | **80.01** | **51.40** | 51.93 | **61.12** |
| Mistral-7B-v0.3 | Task Arithmetic | 57.72 | 42.40 | 50.72 | 50.28 |
| | DARE-Linear | 57.26 | 41.40 | 50.42 | 49.69 |
| | TIES-Merging | **72.08** | 36.40 | 51.86 | 53.45 |
| | DARE-TIES | 57.84 | 43.00 | 50.19 | 50.34 |
| | DELLA-Merging | 57.58 | 42.40 | 51.25 | 50.41 |
| | Rankmean | 51.14 | 42.20 | 50.87 | 48.07 |
| | Evo | 60.66 | **51.93** | 41.80 | 51.47 |
| | PSO-Merging | 71.33 | 41.20 | **53.53** | **55.35** |

Evaluation (GLUE) benchmark (Wang et al., 2018): CoLA, MNLI, MRPC, QNLI, QQP, RTE, SST-2, and STSB. The expert models were sourced from HuggingFace.[1] For evaluation, we report Spearman's $\rho$ for STSB and exact match accuracy for the other tasks.

**Llama-2-13B, Llama-3-8B, and Mistral-7B-v0.3 Experiments**   In accordance with the experimental settings of prior research (Yu et al., 2024a), we conducted experiments on merging three specialized experts: an instruction-following expert, a mathematical reasoning expert, and a code-generating expert. For Llama-2-13B, the three experts were WizardLM-13B-v1.2, WizardMath-13B-v1.0, and Llama-2-13B-Code-Alpaca. For Llama-3-8B and Mistral-7B-v0.3, we trained corresponding experts tailored to each base model. Detailed training procedures are described in Appendix A. For evaluation, we assess instruction-following ability using the win rate on AlpacaEval (Li et al., 2023b), mathematical reasoning ability using zero-shot accuracy on GSM8K (Cobbe et al., 2021), and code-generating ability using `pass@1` on MBPP (Austin et al., 2021). We use Llama-3.1-70B under the Ollama[2] framework as the judge for the AlpacaEval task. We use xFinder[3] (Yu et al., 2024b) to extract the answer for the GSM8K task.

## 3.3 EXPERIMENTAL RESULTS

In the Flan-T5-Base experiment setup, we randomly selected 50 samples from the training set of each task to form the optimization set, which was used to calculate the fitness. The evaluation results are summarized in Table 1. Remarkably, our method outperforms all baseline methods significantly on the MNLI task and demonstrates a clear advantage in average score across all tasks.

For the experiments with Llama-2-13B, Llama-3-8B, and Mistral-7B-v0.3, the dataset for each task was partitioned into an optimization set and a test set, with a 1:10 ratio. Comprehensive dataset statistics are provided in Appendix B. The evaluation results are shown in Tables 2. Notably, our method demonstrates a substantial improvement over all baseline approaches, achieving the highest

---

[1] https://huggingface.co/collections/tanganke/flan-t5-base-models-fine-tuned-on-glue-benchmark-664f30d7966303d9a0a90bb6

[2] https://ollama.com/

[3] https://huggingface.co/IAAR-Shanghai/xFinder-qwen1505

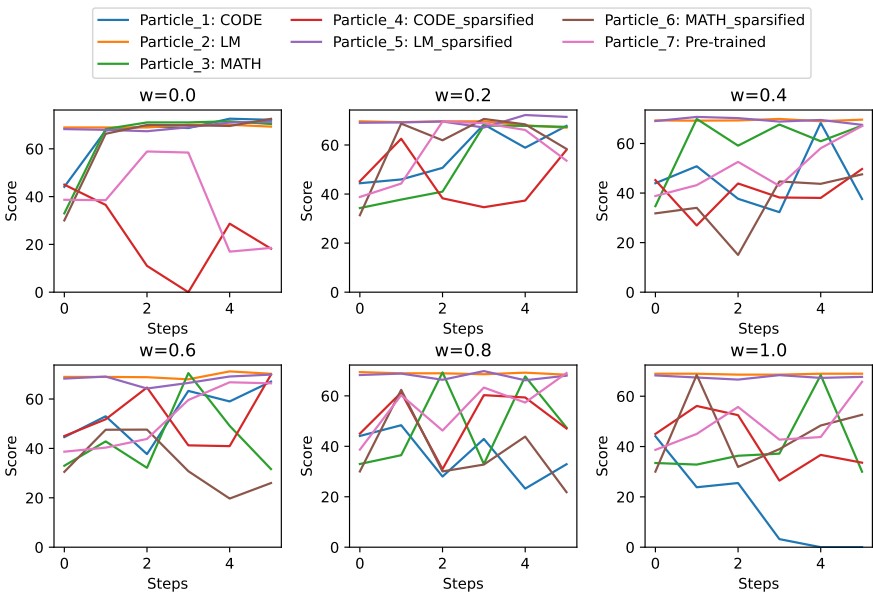

Figure 2: The score variations on the optimization set for all particles with different $w$ values. The legend indicates the expert used to initialize each particle and specifies whether the expert has been sparsified. All lines represent scores on the optimization set. **CODE** denotes the code-generation expert, **LM** refers to the instruction-following expert, and **MATH** represents the mathematical reasoning expert.

Table 3: Multitask performance when merging experts based on Llama-3-8B of four tasks.

| Method | AlpacaEval | MBPP | GSM8K | SciQ | AVG |
|---|---|---|---|---|---|
| Task Arithmetic | 50.63 | 50.20 | 51.86 | 82.20 | 58.72 |
| DARE-Linear | 61.19 | 37.00 | **55.12** | 79.30 | 58.15 |
| TIES-Merging | 53.37 | 49.00 | 54.13 | 82.60 | 59.78 |
| DARE-TIES | 50.49 | 49.60 | 53.22 | 82.80 | 59.03 |
| DELLA-Merging | 51.60 | 49.20 | 53.15 | 82.30 | 59.06 |
| Rankmean | 35.62 | 48.20 | 47.23 | 72.30 | 50.84 |
| Evo | 55.74 | 49.60 | 50.87 | **82.90** | 59.78 |
| PSO-Merging | **80.89** | **50.60** | 49.58 | 76.80 | **64.47** |

average score across all experimental settings. Due to the considerable memory demands associated with gradient computations (Adamerging, Fisher-Merging) or the necessity of retaining intermediate activations (RegMean), precluding their effective application with large-scale models like those explored here, we did not compare against these baselines.

## 4 ANALYSIS

In this section, we explore the impact of some hyper-parameters in our method. Additionally, we validated the effectiveness of our method in scenarios involving the fusion of more experts. All the analysis experiments were conducted under the Llama-3-8B experimental settings.

### 4.1 IMPACT OF THE MOMENTUM COEFFICIENT $w$

We present the optimization process for different choices of $w$ in Figure 2. When $w = 0.0$, most particles converge to high scores, but two particles remain unoptimized. As $w$ increases beyond 0.2, almost all particles fail to optimize, with the situation worsening as $w$ grows larger. Notably, when

Table 4: Multitask performances when setting $\Theta_{\text{initial}} = \Theta$ and $\Theta_{\text{initial}} = \widetilde{\Theta}$.

| Method | AlpacaEval | MBPP | GSM8K | AVG |
|---|---|---|---|---|
| PSO-Merging($\Theta_{\text{initial}} = \Theta$, 3 particles) | 80.85 | 49.80 | 47.84 | 59.50 |
| PSO-Merging($\Theta_{\text{initial}} = \widetilde{\Theta}$, 3 particles) | **81.46** | 50.00 | 50.27 | 60.57 |
| PSO-Merging($\Theta_{\text{initial}} = \Theta \cup \widetilde{\Theta} \cup \{\boldsymbol{\theta}_0\}$, 7 particles) | 80.01 | **51.40** | **51.93** | **61.12** |

$w = 0.2$, all particles successfully converge to comparably high scores, indicating that setting the momentum parameter $w = 0.2$ is a reasonable choice.

## 4.2 PERFORMANCE IN MERGING MORE EXPERTS

We conducted the four-task experiment to explore the performance of PSO-Merging when merging more experts. We incorporated an additional task, SciQ (Welbl et al., 2017) in this experiment. The results are presented in Table 3, demonstrating that PSO-Merging outperforms all methods when merging four experts.

## 4.3 CONVERGENCE BEHAVIOR OF PSO-MERGING

In this section, we investigate the convergence behavior of PSO-Merging. We present in Figure 3 the variation in the fitness scores of the seven particles in the optimization set over 40 optimization steps. Additionally, we illustrate the change in the fitness score of the global best particle on the test set throughout the optimization process. The plot demonstrates that all particles converge rapidly within 10 steps on the optimization set, with the majority converging within the first 5 steps. Notably, PSO-Merging achieves satisfactory performance on the test set within just 5 steps.

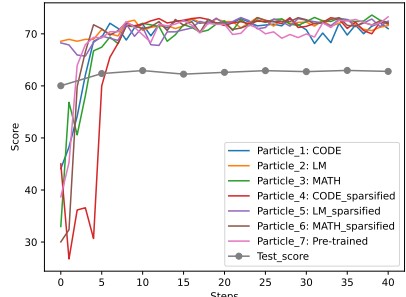

Figure 3: The score variations on both the optimization set and the test set over the course of 40 optimization steps. **Test_score** denotes the score of the global best particle on the test set. All other lines represent the scores of different particles on the optimization set.

## 4.4 EFFECT OF NUMBER OF PARTICLES

To validate the impact of the number of particles in PSO-Merging and the effect of the sparsification mechanism, we conducted experiments using two different configurations: $\Theta_{\text{initial}} = \Theta$ and $\Theta_{\text{initial}} = \widetilde{\Theta}$. The results, presented in Table 4, show that merging only the original experts yields the lowest score while merging only the sparsified experts achieves a higher score. This suggests that the sparsification mechanism effectively reduces parameter conflicts between the models. Furthermore, when $\Theta_{\text{initial}} = \Theta \cup \widetilde{\Theta} \cup \{\boldsymbol{\theta}_0\}$, the score is the highest, indicating that a larger number of particles facilitates the creation of a better-merged model.

## 4.5 EFFICIENCY COMPARED WITH GRADIENT-BASED METHODS

Fisher Merging necessitates computing per-example gradients across all parameters for each expert, incurring a memory cost comparable to training on N examples (around 28GB per 7B model), making it impractical for merging multiple large models. RegMean, requiring the retention of intermediate activations for all merging models during optimization, also presents a significant memory overhead. Similarly, Adamerging, while training only merging weights, requires loading all models simultaneously (e.g., 42GB for three 7B models). In contrast, PSO-Merging operates solely on inference, requiring less memory, exemplified by its 14GB requirement in the same three 7B model scenario.

## 5 RELATED WORK

**Model Merging**    In recent years, model merging has gained significant attention as a versatile approach in machine learning research. Lu et al. (2024) categorizes current studies on model merging into two main directions: merging to achieve a relatively optimal solution (M-ROS) and merging to enhance multitask capability (M-MTC). Our work focuses on M-MTC, aiming at constructing multitask models, and we provide an overview of related studies in this scenario. These methods are categorized into data-independent methods and data-guided methods.

For data-independent methods, Ilharco et al. (2023) introduced the concept of a *Task Vector*, defined as the difference between the parameters of a post-trained model and its corresponding pre-trained model. By multiplying these task vectors with the merging weights and summing them, a merged task vector is obtained. This vector, when combined with the pre-trained parameters, produces the final merged model. Building on this framework, TIES-Merging (Yadav et al., 2023) improves the process by pruning low-magnitude parameters and resolving sign disagreements prior to merging, thereby enhancing its effectiveness. To address parameter conflicts among task vectors, Yu et al. (2024a) proposed DARE, a method that randomly sparsifies and rescales task vectors to reduce task vector redundancy. Alternatively, Deep et al. (2024) introduced DELLA-Merging, which replaces the pruning mechanism of TIES-Merging by employing a probabilistic distribution based on the magnitude rank of task vector parameters. Rankmean (Perin et al., 2024) computes module-specific merging weights for each expert model based on magnitude ranks.

For data-guided methods, Fisher Merging (Matena & Raffel, 2022) uses some labeled data for each task to estimate the diagonal approximate Fisher matrix, which is treated as the importance of the task-specific expert. Then the diagonal approximate Fisher matrix is applied to merge the models as the merging weights. Adamerging (Yang et al., 2023) treats the merging weights as trainable parameters directly, using entropy minimization on unlabeled test samples as a surrogate objective function to optimize the merging weights. To eliminate the need to calculate the gradients, Akiba et al. (2024) propose to use CMA-ES to search for optimal merging weights. However, it requires sampling within the solution space, where many samples are discarded in each iteration because of poor evaluation scores. Consequently, it demands a large number of iterations to obtain satisfactory results, which makes it inefficient. Model Swarms (Feng et al., 2024) is another iterative model merging method.

**Particle Swarm Optimization**    Particle Swarm Optimization (PSO) (Kennedy & Eberhart, 1995) is a stochastic optimization algorithm inspired by the collective intelligence observed in natural phenomena such as bird flocking and fish schooling. In PSO, a population of particles represents potential solutions, each navigating the search space by adjusting its position based on its own experiences and the best solutions discovered by the entire swarm. This cooperative mechanism allows particles to efficiently explore the search space, while also honing in on regions of interest through shared information. The method excels at balancing exploration and exploitation, enabling rapid convergence to optimal or near-optimal solutions.

## 6 CONCLUSION

In this work, we introduce PSO-Merging, a novel model merging method that adapts traditional Particle Swarm Optimization (PSO) to the model merging scenario. We first demonstrate the strong applicability of PSO for model merging tasks. To further enhance its effectiveness, we incorporate a widely used sparsification mechanism, which mitigates parameter conflicts and allows the utilization of a larger number of linearly independent particles. Building on these insights, PSO-Merging leverages task-specific data to produce merged models with superior performance. We also provide an intuitive explanation of its effectiveness. Extensive experiments under various settings show that PSO-Merging achieves effective merging of multiple expert models and consistently outperforms baseline methods. Additionally, our analysis explores the influence of key hyperparameters and confirms the potential of PSO-Merging in scenarios involving the merging of a greater number of experts.

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

## A   TRAINING DETAILS FOR EXPERTS BASED ON LLAMA-3-8B AND MISTRAL-7B-V0.3

In this section, we describe the training processes used to develop four domain-specific experts based on Llama-3-8B and Mistral-7B-v0.3. All training was conducted on 4 RTX 3090 GPUs. The training hyperparameters include a gradient accumulation step size of 32, a per-device training batch size of 1, and a learning rate of $5 \times 10^{-6}$. The datasets used for training, the number of epochs, and the corresponding expert models are outlined below. Our training was conducted using the Hugging Face Transformers framework.

**Instruction-following expert**   We fine-tuned Llama-3-8B and Mistral-7B-v0.3 on the Infinity-Instruct[4] dataset for 1 epoch to create the instruction-following expert. This dataset provides high-quality instruction-response pairs, enabling the model to excel in general instruction-following tasks.

**Mathematical reasoning expert**   To develop the mathematical reasoning expert, we fine-tuned Llama-3-8B and Mistral-7B-v0.3 on the MathInstruct[5] dataset for 1 epoch. This dataset focuses on math-related problems and solutions, allowing the model to specialize in solving mathematical reasoning tasks.

---

[4] `https://huggingface.co/datasets/BAAI/Infinity-Instruct`
[5] `https://huggingface.co/datasets/TIGER-Lab/MathInstruct`

Table 5: The statistics of datasets used in our experiments. The label in parentheses indicates which split the current split is derived from in the source data.

| Dataset | Training Split | Optimization Split | Testing Split |
|---|---|---|---|
| Infinity-Instruct | 659,808(train) | - | - |
| AlpacaEval | - | 73(eval) | 732(eval) |
| MathInstruct | 262,039(train) | - | - |
| GSM8K | - | 131(train) | 1,319(test) |
| CodeAlpacaPython | 8,477(train) | - | - |
| MBPP | - | 50(train) | 500(test) |
| SciQ | 11,679(train) | 100(validation) | 1,000(test) |

**Code-generating expert**   The code-generating expert was obtained by fine-tuning Llama-3-8B and Mistral-7B-v0.3 on the CodeAlpacaPython[6] dataset for 5 epochs. This dataset contains Python-specific programming problems and solutions, which help the model specialize in code generation.

**Science exam question-answering expert**   .   For science-related question answering, we fine-tuned Llama-3-8B and Mistral-7B-v0.3 on the SciQ[7] dataset for 5 epochs. The dataset consists of multiple-choice science questions, enabling the model to perform well in science exam scenarios.

## B   DATASET STATISTICS

The data statistics for training, optimization, and testing are listed in Table 5.

---

[6]https://huggingface.co/datasets/Abzu/CodeAlpacaPython
[7]https://huggingface.co/datasets/allenai/sciq

