# OpenReview forum: "PSO-Merging: Merging Models Based on Particle Swarm Optimization"
_ICLR.cc/2026/Conference — Submitted to ICLR 2026_

### Official Review · Reviewer_naRy · 2025-10-27

**Soundness:** 2
**Presentation:** 2
**Contribution:** 1
**Rating:** 2
**Confidence:** 5

**Summary:**

The paper introduces **PSO-Merging**, a  data-driven method for model merging that leverages **Particle Swarm Optimization (PSO)** to create multitask models. Model merging is a post-training strategy that constructs new models by combining previously fine-tuned models that share a common pretrained backbone. **PSO-Merging** positions itself as an intermediate approach between data-free model merging and gradient-based post-training techniques for building such models.

The method is straightforward: initialize a population by selecting *n* fine-tuned models that share the same pretrained model and their corresponding sparsifications, then apply *S* update steps to the model population by (1) evaluating the current models and (2) performing a sort of linear combination with the best model from the history and the current best model. The top-performing model is retained as the final one.

Experiments conducted across various large language models (Flan-T5, LLaMA, Mistral) demonstrate that **PSO-Merging** outperforms data-free baselines and an evolutionary algorithm poorly described (and possibly employed).

**Strengths:**

**Empirical Setting is Good** : The paper manage to tackle the problem of model merging in a realistic scenario; the model chosen are common LLMs, use by practitioner on Hugging Face.

**Weaknesses:**

The work lacks novelty. In particular, the paper does not cite [4], where model merging is proposed as a core element of a swarm dynamic in a very similar way. The current paper essentially applies PSO as a plug-and-play optimization framework around existing baselines, reusing the sparsification technique directly from DARE [1]. As a result, the proposed method appears to be a relatively minor wrapper on top of established techniques, following the same general scheme as [4].

While PSO has not been explored in exactly this context of model merging, the approach (1) lacks clear theoretical motivation, (2) produces results that are not particularly strong, and (3) is conceptually similar to prior work [4]. Moreover, PSO itself offers no obvious advantages over other black-box optimizers, and evolutionary model merging has already been studied extensively [2, 3, 4] with stronger empirical results.

The experimental results also raise several concerns. Baseline evaluations for the individual fine-tuned models used for initialization are missing. The comparison between EVO and PSO is unclear: for fairness, the EVO result should correspond to the best solution found during optimization rather than the final iteration, but the paper does not specify if this is the case. In Table 3, PSO achieves only marginal improvements over DARE-TIES for LLaMA-3, contrasting with state-of-the-art data-driven methods [2, 3] that typically report double-digit gains. Overall, the evaluation pipeline appears incomplete, and the reported improvements remain minor.

The paper’s structure could also be improved: the baseline section reads more like related work. Consider merging it with Section 5 (Related Work) or moving it earlier to avoid redundancy.
For these reasons (lack of novelty, and flawed evaluation), I suggest rejecting the paper.

[1] Yu, Le, et al. "Language models are super mario: Absorbing abilities from homologous models as a free lunch." *Forty-first International Conference on Machine Learning*. (ICML) 2024.

[2] Akiba, Takuya, et al. "Evolutionary optimization of model merging recipes." *Nature Machine Intelligence* (2025): 1-10.

[3] Mencattini, Tommaso, et al. "MERGE3: Efficient Evolutionary Merging on Consumer-grade GPUs." Forty-Second International Conference on Machine Learning (ICML) (2025).

[4] Feng, Shangbin, Zifeng Wang, Yike Wang, Sayna Ebrahimi, Hamid Palangi, Lesly Miculicich, Achin Kulshrestha, Nathalie Rauschmayr, Yejin Choi, Yulia Tsvetkov, Chen-Yu Lee, and Tomas Pfister. “Model Swarms: Collaborative Search to Adapt LLM Experts via Swarm Intelligence.” *Forty-Second International Conference on Machine Learning (ICML)*. 2025.

**Questions:**

**Q1)** For a fair comparison between EVO and PSO, the solution returned by EVO should not be the last one, but the best one. Is this the case?

**Q2)** Was at least one of the elements of EVO's initial population the model merged with the hyperparameter chosen for the reported DARE_TIES? Did EVO share the initial population with PSO?

---

### Official Review · Reviewer_MWhe · 2025-10-28

**Soundness:** 3
**Presentation:** 1
**Contribution:** 1
**Rating:** 0
**Confidence:** 4

**Summary:**

The authors introduce PSO-Merging, a model merging algorithm that optimizes a population composed of the pretrained model, fine-tuned experts, and sparsified versions of these experts obtained through a DARE-based sparsification mechanism. The framework employs a Particle Swarm Optimization (PSO) process, where each particle represents a candidate merged model. At each iteration, particles update their positions based on their personal best and global best fitness scores, computed as the average multitask performance. The final merged model corresponds to the global best particle after optimization. Experiments across multiple model architectures (Flan-T5-Base, Llama-2-13B, Llama-3-8B, and Mistral-7B-v0.3) show that PSO-Merging consistently achieves competitive or superior results compared to baselines such as Task Arithmetic[1], DARE[2], TIES[3], RankMean[4], Evolutionary Merging[5], and DELLA-Merging[6], while maintaining efficiency through its gradient-free nature.

**Strengths:**

- The optimization algorithm is well-defined, gradient-free, and demonstrates rapid convergence in practice.
- Extensive experiments across diverse model families and benchmarks, including GLUE and multitask setups, showing consistent improvements over multiple baselines.
- The paper provides ablations on the momentum parameter, number of particles, and sparsification mechanism, supporting the method’s stability.

**Weaknesses:**

1. The relationship between PSO-Merging and prior swarm-based approaches (e.g., Model Swarms[7]) is mentioned but not discussed in depth, making it unclear what the methodological novelty is.

2. Training details of the expert models (especially for Llama-3 and Mistral) are not fully disclosed (the optimizer is not specified), limiting the reproducibility of the experiments.

3. The presentation suffers from grammatical issues, redundant explanations, and hard to parse notation (lines 166-167), particularly in the methodology section.
	- “…our novel model merging method based on the PSO.” (lines **84-85**) → (drop “the”)
	- “An overview of PSO-Merging is demonstrated as Figure 1.” (lines **97-98**) → “An overview of PSO-Merging is shown in Figure 1.”
	- “So our initial solution set can be represented as …” (lines **150-151**) → “Thus, the initial solution set is…” (or simply “The initial solution set is …”)

4. Some baselines (e.g., Adamerging, Fisher-Merging, RegMean) are excluded in Table 3 due to computational constraints, which slightly limits the empirical comparison.

**Questions:**

1. Could the authors clarify in detail how PSO-Merging differs algorithmically from **Model Swarms**, beyond the initialization of particles from sparsified experts?

2. Could the authors report **wall-clock time or FLOP estimates** for PSO-Merging compared to other baselines to better quantify the claimed efficiency advantages?

### References
[1] Ilharco, G., Ribeiro, M., Wortsman, M., Gururangan, S., Schmidt, L., Hajishirzi, H., & Farhadi, A. (2023). The Eleventh International Conference on Learning Representations.

[2] Le Yu, Bowen Yu, Haiyang Yu, Fei Huang, and Yongbin Li. Language models are super mario: Absorbing abilities from homologous models as a free lunch (2024). In Forty-first International Conference on Machine Learning.

[3] Prateek Yadav, Derek Tam, Leshem Choshen, Colin A Raffel, and Mohit Bansal. TIES-merging: Resolving interference when merging models (2023). In A. Oh, T. Naumann, A. Globerson, K. Saenko, M. Hardt, and S. Levine (eds.), Advances in Neural Information Processing Systems, volume 36, pp. 7093–7115.

[4] Gabriel Perin, Xuxi Chen, Shusen Liu, Bhavya Kailkhura, Zhangyang Wang, and Brian Gallagher. Rankmean: Module-level importance score for merging fine-tuned LLM models (2024). In Findings of the Association for Computational Linguistics: ACL 2024, pp. 1776–1782.

[5] Akiba, T., Shing, M., Tang, Y. _et al._ Evolutionary optimization of model merging recipes (2025). _Nat Mach Intell_ **7**, 195–204.

[6] Pala Tej Deep, Rishabh Bhardwaj, and Soujanya Poria. Della-merging: Reducing interference in model merging through magnitude-based sampling. _ArXiv, abs/2406.11617_.

[7] Feng, S., Wang, Z., Wang, Y., Ebrahimi, S., Palangi, H., Miculicich, L., Kulshrestha, A., Rauschmayr, N., Choi, Y., Tsvetkov, Y., Lee, C., & Pfister, T. (2024). Model Swarms: Collaborative Search to Adapt LLM Experts via Swarm Intelligence. _ArXiv, abs/2410.11163_.

---

### Official Review · Reviewer_YvRH · 2025-10-31

**Soundness:** 3
**Presentation:** 2
**Contribution:** 3
**Rating:** 4
**Confidence:** 4

**Summary:**

The paper introduces PSO‑Merging, a data‑driven but gradient‑free method for merging several task specific experts into a single multitask model. It borrows the idea from PSO, modeling the model collection, which includes the pretrained model, expert model, and sparsified expert model, as particles, and updates them according to the average task score. The global best particle is taken as the merged model. They evaluate on LLaMA‑2, LLaMA‑3, Mistral‑7B, and Flan‑T5, and achieve improved performance over various baselines.

**Strengths:**

- Broad experiments across Flan-T5, LLaMA, and Mistral, covering GLUE, MATH, Instruct, and Code, compared with various methods like Task Arithmetic, DARE, TIES, and AdaMerging.
- The PSO idea is interesting: it treats the entire model as a particle and can wrap around existing experts.
- Can only use small labeled optimization sets: for GLUE 50 training samples per task, for LLaMA/Mistral a 1:10  split.

**Weaknesses:**

- W1: This method requires access to an external judge and task metrics during the swarm process. With $N$ tasks, the swarm has $2N+1$ particles (experts + sparsified experts + base). With $S$ steps (main results: $S=5$ for LLaMA/Mistral; $S=50$ for Flan‑T5), the number of scorings for each metric is $(2N+1) \times S$, which is costly when the judge is a large LLM or the experts are large LLMs.

- W2: As this method requires test metric results, it risks data leakage and is unfair to compare with merging methods that require no data (e.g., Task Arithmetic) or only unlabeled data (e.g., adaMerging). The essence of this method is to search over different evaluation metrics to yield a better result.

- W3: The fitness is a simple average of heterogeneous metrics, which is unreliable. Fitness uses a plain mean across tasks; in Table 2, typical per-task scales differ (e.g., AlpacaEval ≈ 73–82, MBPP ≈ 16–51, GSM8K ≈ 29–57 depending on the model), so direct averaging implicitly down-weights the higher-range metric or up-weights the lower-range one.

- W4: The improvements are marginal: Table 1 shows outperformance over other baselines on only 3 out of 8 tasks; Table 2 shows outperformance on only 1 out of 3 tasks for both Mistral and LLaMA2; Table 3 shows outperformance on only 2 out of 4 tasks.

- W5: The maximum number of tasks involved is four; the number of particles grows with the number of experts ($2N+1$), so evaluating all particles at each step becomes costly for larger $N$. There is no $N$-scaling study. Table 4 explicitly shows that performance improves when increasing the particle count from 3 to 7, implying that more particles matter, which will be costly.

- W6: It seems empirical to include DARE‑sparsified experts in the particle initialization; many hyperparameters are fixed ($p=0.8$; $c_1=c_2=2$; $w=0.2$); sensitivity analysis is limited (only $w$ is studied).

**Questions:**

See Weakness

---

### Official Review · Reviewer_cTXw · 2025-11-01

**Soundness:** 2
**Presentation:** 2
**Contribution:** 2
**Rating:** 4
**Confidence:** 3

**Summary:**

This paper proposes PSO-Merging, a novel data-driven merging method based on the Particle Swarm Optimization (PSO). To address the limitations of costly gradient-based merging and the poor effectiveness of existing gradient-free methods. The PSO-Merging initializes the particle swarm with a pre-trained model, expert models, and sparsified expert models. Then perform multiple iterations, with the final global best particle serving as the merged model. Experimental results show that the PSO-Merging achieves higher average scores than the comparison methods in some settings.

**Strengths:**

+ This paper proposes a novel model merging method, PSO-Merging. Experimental results show that PSO-Merging outperforms baseline methods in terms of average performance.

+ Compared with gradient-based approaches, it requires less memory, making it more practical for large-scale model merging.

**Weaknesses:**

- Generalization concerns remain. Although PSO-Merging achieves the highest average performance, these improvements are often driven by strong results on only a subset of tasks. In many cases, the method performs worse than some baselines at the per-task level (e.g., compared with DELLA-Merging in Table 1). This raises concerns about whether the approach consistently generalizes across diverse task types.

- Efficiency evaluation is insufficient and lacks clarity. A key motivation of the work is to address the high computational cost of gradient-based methods. However, the paper provides only brief and qualitative discussion of computational efficiency, and only compares memory usage.

- Lack of interpretation and insight into experimental outcomes. The experimental section mainly reports numerical results but does not explain why PSO-Merging performs better on some tasks yet worse on others.

- Minor weakness: It should be ""Table 2"" in line 319, rather than ""Tables 2""."

**Questions:**

Q1: Could the authors provide more detailed experimental settings and additional efficiency comparisons, including metrics beyond memory usage?

Q2: Could the authors explain why PSO-Merging performs better on certain tasks but worse on others, and analyze why other methods underperform under the experimental configurations used in this paper?"

**Details Of Ethics Concerns:**

No ethical issues identified.

---

### Meta-Review · Area_Chair_wxPm · 2026-01-04

**Summary:**

The paper had one strong reject, one reject, and two weak rejects. The authors **did not submit a rebuttal**. Given this is such a clear case, I recommend rejection.

**Reviewer Concerns:**

None of the concerns raised were addressed by the rebuttal, as there was no rebuttal.

**Reviewer Scores:**

Reviewers would not have changed their scores if they had been able to participate fully in the discussion.

---

### Decision · Program_Chairs · 2026-01-26

Reject